# Prevalence and incidence of physical health conditions in people with intellectual disability – a systematic review

**Peiwen Liao** [1]*, **Claire Vajdic**[2], **Julian Trollor**[1], **Simone Reppermund**[1]

**1** Department of Developmental Disability Neuropsychiatry, University of New South Wales, Sydney, Australia, **2** Centre for Big Data Research in Health, University of New South Wales, Sydney, Australia

* Peiwen.liao@student.unsw.edu.au

## Abstract

### Objective

To synthesize evidence on the prevalence and incidence of physical health conditions in people with intellectual disability (ID).

### Methods

We searched Medline, PsycInfo, and Embase for eligible studies and extracted the prevalence, incidence, and risk of physical health conditions in people with ID.

### Results

Of 131 eligible studies, we synthesized results from 77 moderate- to high-quality studies, which was mainly limited to high-income countries. The highest prevalence estimates were observed for epilepsy, ear and eye disorders, cerebral palsy, obesity, osteoporosis, congenital heart defects, and thyroid disorders. Some conditions were more common in people with a genetic syndrome. Compared with the general population, many health conditions occur more frequently among people with ID, including asthma and diabetes, while some conditions such as non-congenital circulatory diseases and solid cancers occur at the same or lower rate. The latter associations may reflect under-detection.

### Conclusions

People with ID have a health profile more complex than previously known. There is a pressing need for targeted, evidence-informed population health initiatives including preventative programs for this population.

**Data Availability Statement:** All relevant data are within the manuscript and its Supporting Information files.

**Funding:** This work was supported by National Health and Medical Research Council (NHMRC;

Grant name: Partnership Project APP1056128,
Project Grant APP1123033); and Scientia PhD
Scholarship. Authors who received each grant:
Scientia PhD scholarship: PL (URL to sponsor's
websites: https://www.scientia.unsw.edu.au/)
NHMRC grant APP 1056128: JT; NHMRC grant
APP 1123033: JT, CV, SR (URL to sponsor's
websites: https://www.nhmrc.gov.au/funding) The
funders had no role in study design, data collection
and analysis, decision to publish, or preparation of
the manuscript.

**Competing interests:** The authors have declared
that no competing interests exist.

## Introduction

People with intellectual disability (ID), defined by cognitive and adaptive-functioning impairments with onset during childhood, constitute about 1% of the global population [1]. They experience significant health inequalities and barriers to effective healthcare and are more likely to die prematurely and from potentially avoidable causes than the general population [2, 3]. The excess deaths amenable to healthcare interventions suggest inadequate recognition of health needs specific to people with ID in healthcare policy and systems [4, 5]. Effective and equitable healthcare requires a comprehensive understanding of the epidemiology of health conditions in this population.

The evidence on psychiatric conditions in people with ID is extensive [6–8], whereas that for physical health conditions is relatively sparse [9]. Compared to the general population, people with ID have a higher prevalence of physical conditions [10], especially neurological disorders, sensory impairments, obesity, constipation, and congenital malformation [11–15]. In contrast, they are less likely to have solid cancers [16, 17]. Low awareness of the disease epidemiology may expose this population to underdiagnosis, misdiagnosis, inappropriate pharmaceutical interventions, and missed opportunities for preventative healthcare.

Previous reviews mainly examined single health conditions [11, 18–32], such as epilepsy, or a specific sub-population (e.g., people with Down Syndrome (DS)) [15, 33]. Only one review included risk estimates relative to the general population [34]. Importantly, few reviews required representative populations [35], and none, to our knowledge, excluded studies potentially subject to bias.

This systematic review aimed to synthesise the population-based prevalence and incidence of physical health conditions in people with ID and compared with the general population. We sought to better inform approaches to recognition and management of physical health conditions and the inclusion of this vulnerable population in preventative health programs and policy.

## Methods

This systematic review follows the Preferred Reporting Items for Systematic Reviews and Meta-Analyses (PRISMA) guidelines and is registered with PROSPERO (CRD42019132214) [36]. Throughout this paper, ID, if unspecified, refers to ID of any aetiology.

### Search strategy

We searched MEDLINE, PsycINFO, and EMBASE. Search terms included ID, physical disorders, and prevalence/incidence or related terms. Each search term was translated into Medical Subject Headings where possible. Details are provided in S5 File. Initially, journal articles published in English on or before 28 February 2020 were included; we also searched reference lists of included articles. The search was updated on 11 May 2021.

### Study selection

Two reviewers independently screened titles and abstracts to identify original peer-reviewed studies reporting prevalence or incidence of physical disorders in people with ID or genetic syndromes invariably related to ID, such as DS.

Two reviewers then screened the full texts independently. Studies were eligible if they were quantitative, observational studies reporting prevalence or incidence, or values that permitted their calculation. Samples were considered population-based if they were derived from a representative sampling frame such as multiple disability services, disability services combined with multiple healthcare organisations, birth cohorts/registries, national or regional medical

registries of people with ID, or both mainstream and special schools. Purely hospital-based samples were excluded unless the authors justified representativeness or where health initiatives are known to prompt proactive identification and registration of people with ID (i.e., UK GP registries). Studies with voluntary/convenience participants or people with additional specific health conditions were excluded, as they were not likely to represent the whole ID population and were at risk of selection bias. To maximise comparability across studies and generalisability of findings, we excluded conditions that could not be mapped to the 10th revision of the International Statistical Classification of Diseases and Related Health Problems (ICD-10).

We only included the study with the highest quality rating if multiple studies had the same or largely overlapping study populations; when such studies received the same quality rating, only the most recent was included.

### Data extraction and quality assessment

We developed a standardised data extraction form for study characteristics, participants (including people with ID and controls), and outcomes. Data were extracted by the primary author and cross-checked by the second reviewer. Disagreements were resolved through consensus and discussion with a third reviewer if necessary.

We adapted the Newcastle-Ottawa Scale (NOS), a validated and widely used tool, to assess the quality of included studies (S1 and S2 Tables in S1 File) [37]. The NOS framework was unchanged, the adaptations made the NOS fit-for-purpose for our study design. Two reviewers independently classified studies as low, medium, or high quality (S3 Table in S1 File).

### Data synthesis

Low quality studies were excluded to maximise evidence validity. Disorders were grouped into ICD-10 disease chapters. We used a narrative analysis approach [38], as heterogeneous samples and methodologies prevented meta-analysis. Prevalence and incidence were summarized separately for people with ID of any aetiology and with genetic causes (e.g., DS). For comparative studies, only age-adjusted estimates (i.e., controlled for age or single age group) were included.

### Patients and public involvement

No patients or the public were involved in any study stage as the research reviewed published data.

## Results

In total, we identified 6976 studies from the database and manual searches after removing duplicates (Fig 1). 871 full texts were assessed for eligibility, and 131 studies published 1970–2021 met the inclusion criteria.

### Study populations

S4 and S5 Tables in S2 File show the study characteristics. Over half were conducted in Europe (n = 77), predominantly the UK (n = 30) and Scandinavia (n = 26). The remaining were mainly from North America and Australasia. No studies were from low-income countries.

The sample size ranged from 24 to 64 008. Sixty-eight studies did not distinguish ID aetiology, while 63 included people only with DS (n = 51), Prader-Willi syndrome (PWS; n = 3), Fragile X syndrome (FXS; n = 3), Rett syndrome (RS; n = 3), or others (n = 3).

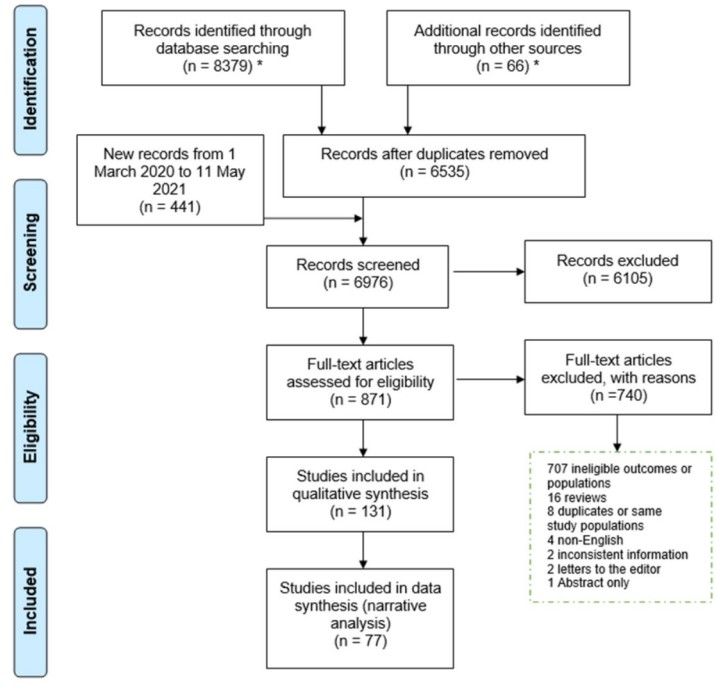

**Fig 1. PRISMA flowchart diagram for study selection.**

Forty-eight studies reported ID severity; most were mild to profound (n = 38). All except 3 studies reported participant age: ≥14 years (n = 56), children and young adults (£29 years) (n = 49), and children and adults (n = 23).

Forty-two studies included comparison populations, either individuals without ID or chromosomal abnormalities, or the general population (S6 and S7 Tables in S2 File). The comparison group(s) ranged in size from 23 to 16 813 290.

## Study quality and methods

The score for each quality assessment item can be found in S8 Table in S3 File; the majority of studies were judged to be of moderate or high quality. Seventy-six studies identified people with ID from national or regional registries including registries linked to birth defect surveillance or primary health care (n = 61), national or regional surveys (n = 7), or birth cohorts (n = 8). Twelve studies diversified their sample sources to make their study populations as representative as possible, with disability and health service users being the most frequent combination. Forty-three studies involved a single source of participants such as disability service users.

ID was ascertained by clinical examination (7.6%), medical records/registry (58.8%), non-health registries such as schools (2.3%), and self- or informant-reports (5.3%). Diagnostic classification systems for ID varied across studies; ICD (n = 25) [13, 16, 39–61], American Association on Intellectual and Developmental Disabilities (n = 5) [62–66], other (n = 16) [10, 67–80], and the remainder did not report the diagnostic guideline used (n = 85). The methods to ascertain the health conditions included clinical examination (11.1%), medical records/registry (54.3%), and self- or informant-reports (29.7%).

Seventy-seven studies were of moderate to high quality and entered the final evidence synthesis. Over half of these studies were conducted in Scandinavia and the UK. Study

populations included children and adults with ID of any aetiology (n = 31) and with a specific genetic syndrome (n = 46). Twenty-two studies reported ID severity, and most covered all severities, from mild to profound. Study populations were mostly derived from multiple sources or comprehensive healthcare registries (i.e., UK GP registries and hospital records to form a birth cohort). Only a few studies were solely based on disability service records. ID diagnosis was ascertained from medical records (n = 61), clinical assessments (n = 7), or was not reported (n = 9). Few studies reported the diagnostic classification system; only 19 used the ICD. Most studies (n = 58) ascertained at least one health outcome from medical records, and 14 studies performed physical examinations. Fig 2 shows the distribution of physical disorders before and after exclusion based on study quality.

Nineteen studies with comparison groups were rated as moderate to high quality. These studies reported odds/prevalence/incidence/standardised incidence ratios (OR/PR/IR/SIR). Comparison groups were mainly from the same registries or surveys. All but one study adjusted for demographic factors or included only children. This study was classified as moderate quality because other methodological features were robust [81], however, the PRs from this study were not included in the evidence synthesis.

## Summary of prevalence and incidence of physical health conditions

S9–S12 Tables in S4 File display results from the 77 moderate- to high-quality studies by ICD-10 disease chapter and measure type. A wide range of physical health conditions was identified, spanning 15 ICD-10 disease chapters. The number of studies varied strikingly across ICD-10 disease chapters (Fig 2).

## The most prevalent physical health conditions in people with ID of any aetiology

Epilepsy (9.0%-51.8%) [10, 44, 45, 51–53, 58, 65–69, 72, 81–87], visual impairment (3.2%-47.0%) [10, 44, 52, 65, 66, 82, 83], hearing loss (1.4%-34.9%) [10, 65, 66, 82, 88], osteoporosis (1.7%-41.0%) [68, 89], obesity/overweight (3.9%-34.8%) [46, 52, 66, 83, 90–92], cerebral palsy (1.0%-28.9%) [65, 67, 69, 72, 82, 84, 86], and microcephaly (20.9%) [69] were the most prevalent disorders in people with ID (Fig 3). Prevalence estimates varied markedly across studies for several disorders, likely due to variation in study population age and ID severity. For example, lower prevalence estimates for obesity were observed in adolescents and relatively young adults than in older populations [66, 91, 92]. For epilepsy and cerebral palsy, the lowest

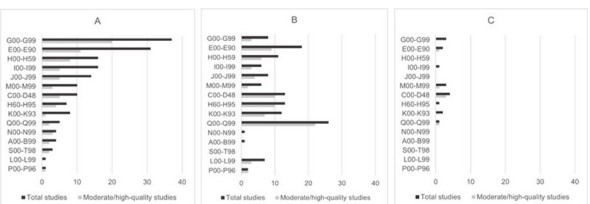

**Fig 2.** Number of studies by ICD-10 disease chapter before and after quality assessment for (A) people with ID of any aetiology, (B) people with Down syndrome, and (C) people with other genetic syndromes. G00-G99: Neurological disorders; E00-E90: Endocrine, nutritional, and metabolic disorders; H00-H59: Eye disorders; I00-I99: Cardiovascular disorders; J00-J99: Respiratory disorders; M00-M99: Musculoskeletal disorders; C00-D48: Neoplasms; H60-H95: Ear disorders; K00-K93: Gastrointestinal disorders; Q00-Q99: Congenital abnormalities; N00-N99: Genitourinary disorders; A00-B99: Infectious disorders; S00-T98: Injuries; L00-L99: Skin disorders; P00-P96: Perinatal-oriented disorders.

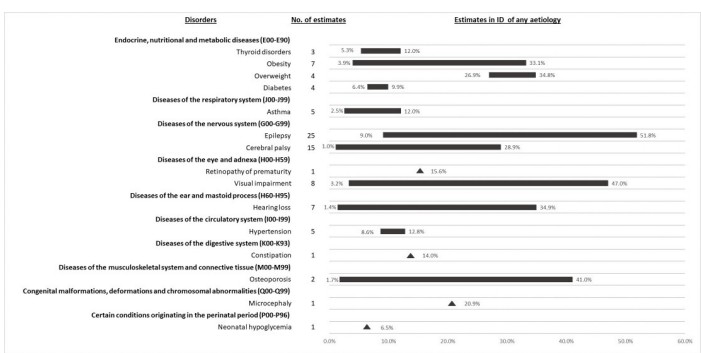

**Fig 3. Prevalence range (%) of 15 most common physical health conditions in people with ID of any aetiology.**

estimates were for older people [58, 84], and the highest estimates for people of all ages and for children with severe to profound ID [67, 82]. Two epilepsy studies observed trends by level of ID severity (12.2%-12.4%-22.8%-59% [45] and 11.3%-12.5%-15.3%-26.3% [65] for mild-moderate-severe-profound respectively). The corresponding prevalence estimates for a cerebral palsy study were 12.2%-14.3%-33.7%-74.6% [65].

**The most prevalent physical health conditions in people with genetic syndromes.** The disease pattern in the whole ID population was similarly in people with DS (Fig 4). Obesity/overweight (25.5%-59.5%) [71, 93, 94], visual impairment including blindness (0.8%-34.9%) [94–96], hearing loss (0.9%-57.4%) [88, 97–103], middle ear infection (30.0%-93.0%) [97, 99, 102, 104, 105], congenital heart defects (unspecified) (14.1%-79.2%) [40, 43, 47, 57, 59, 76, 78, 96, 97, 101, 105–112], cold or influenza (78.7%) [99], thyroid disorders (1.0%-39.0%) [95, 97, 96, 99, 105, 107, 112, 113], common skin diseases (13.0%-23.4%) [96, 113], refractive error (14.4%-29.7%) [99], and lower respiratory tract infection (11.4%-27.0%)[97, 99] were common in people with DS (Fig 4). For hearing and visual impairments, the lower prevalence estimates were found for specific hearing [98, 100, 102, 104] or visual impairments [94, 96], including deafness (0.9%) [101] and blindness (0.8%) [94], and higher prevalence estimates for unspecified impairments. The lowest prevalence of unspecified hearing loss in children with DS was based on parent-reported data [99]. The age of the study populations may partly explain these variations. For example, the highest prevalence of middle ear infection in people with DS was in children aged one (93.0%) and it decreased with increasing age [105]. Thyroid disorders also appeared to be age sensitive, with lower prevalence estimates for children or adolescents [99, 107, 112]. One study reported a higher prevalence of thyroid disorder for participants aged 65–74 (45.5%) than those aged 45–64 (33.6%) [95].

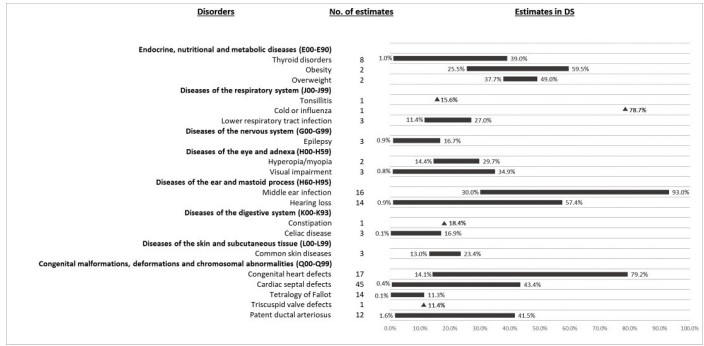

**Fig 4. Prevalence range (%) of 15 most common physical health conditions in people with Down syndrome.**

Prevalence estimates in people with other genetic syndromes are limited. Hypothyroidism, hypogonadism and cryptorchidism were prevalent in children and adolescents with PWS at 24.4%, 93.1% and 87.8%, respectively [70]. Additionally, scoliosis was found in 33.0% of people with Angelman syndrome [114].

**The incidence of physical health conditions in people with ID.** Most studies reporting incidence examined cancer (S10 Table in S4 File). Crude incidence rates for all cancers ranged from 12.8 to 33.1 per 10 000 person-years in people with ID of any aetiology [16, 77], from 6.4 to 12.6 per 10 000 person-years in people with DS [54, 115–117] and 29 per 10 000 person-years in people with PWS [55]. The incidence of solid tumours was low, regardless of ID aetiology, with the highest incidence observed for gastrointestinal cancer in people with mild to moderate ID [16]. There was a high incidence of leukemia in children with DS [115]. The cancer incidence rates were not age-standardised, thus differences in age distributions between ID subgroups may partly explain the observed variation.

Two studies reported the incidence of other conditions. A British study examined injury incidence and found that on average 20 per 100 adults with ID experienced accidental injuries within one year [44]. Another study reported an incidence of 32.5 per 10 000 person-years for celiac disease in people with DS [61].

## Comparisons of physical health conditions between people with ID and the general population

Risk estimates for physical health conditions from 14 ICD-10 disease chapters were identified (S11 and S12 Tables in S4 File).

**People with ID of any aetiology.** People with ID of any aetiology had a significantly elevated risk of 16 diseases. Diabetes (undefined or both type I and II), epilepsy and asthma were consistently reported to be more prevalent [10, 53, 68, 95, 118]. Epilepsy had the highest effect size, ranging from 23.73 to 31.03. Thyroid disorders (OR = 2.36; 95% CI = 2.17, 2.58) [10], constipation (OR = 11.19; 95% CI = 10.97, 12.68) [10], hearing loss (OR = 2.81; 95% CI = 2.59, 3.06) [10], visual impairment (OR = 7.81; 95% CI = 6.86, 8.89) [10], retinopathy of prematurity (PR = 2.79; p < 0.05) [39], injuries (SIR = 1.78; 95% CI = 1.44, 2.17) [44], osteoporosis (PR = 1.84; 95% CI = 1.60, 2.12) [68], migraine (OR = 1.32; 95% CI = 1.02, 1.71) [10], Parkinson's disease (OR = 2.82; 95% CI = 1.95, 4.13) [10], bronchiectasis (OR = 1.68; 95% CI = 1.08, 2.61) [10], and gallbladder cancer (SIR = 2.8; 95% CI = 1.1, 5.8)[16] were reported to occur more frequently in people with ID.

People with ID had a significantly lower risk of some conditions, including diverticular disease (OR = 0.49; 95% CI = 0.25, 0.96) [10], prostate disease (OR = 0.60; 95% CI = 0.44, 0.82) [10], inflammatory arthritis (OR = 0.57; 95% CI = 0.48, 0.67) [10], multiple sclerosis (OR = 0.49; 95% CI = 0.39, 0.63) [10], chronic sinusitis (OR = 0.44; 95% CI = 0.26, 0.62) [10], and cancer (SIR = 0.9; 95% CI = 0.8, 1.0) [16]. Specifically, cancers of the male genital organs (SIR = 0.4; 95% CI = 0.1, 0.8) and urinary tract (SIR = 0.3; 95% CI = 0.1, 0.7) were observed less frequently in people with ID in one study [16].

Risk of coronary heart disease was reduced by 35–56% among adults with ID in the UK (PR = 0.65, 95% CI = 0.57, 0.74; OR = 0.43, 95% CI = 0.37, 0.51), while risk of hypertension (PR = 0.93, 95% CI = 0.89, 0.98; OR = 0.72, 95% CI = 0.66, 0.78), chronic obstructive pulmonary disease (PR = 0.84, 95% CI = 0.71, 0.99; OR = 0.84, 95% CI = 0.73, 0.97) and atrial fibrillation (PR = 0.91, 95% CI = 0.75, 1.09; OR = 0.83, 95% CI = 0.61, 0.98) were modestly decreased [10, 68]. The evidence is mixed for peripheral vascular disease, (OR = 0.44, 95% CI = 0.33, 0.60 [10]; PR = 0.90, 95% CI = 0.69, 1.17) [68], heart failure (PR = 2.26; 95% CI = 1.84, 2.78 [68]; PR = 1.11; 95% CI = 0.89, 1.43 [10]) and chronic kidney disease (PR = 1.64; 95% CI = 1.49,

1.82 [68]; OR = 1.11; 95% CI = 0.93, 1.32 [10]). These risk estimates are from two UK GP practice studies with similar population characteristics and similar methods [10, 68]. One study randomly selected non-ID controls from the same GP practice as each case, matched on age and sex, and thus may have controlled for any practice variation in diagnostic care and the quality and completeness of recording for individuals with and without ID [68].

The published evidence indicated similar risks in people with ID and the general population for HIV, viral hepatitis [10, 119], gastrointestinal diseases (inflammatory bowel disease, irritable bowel syndrome) [10], glaucoma [10], and many types of cancer (mainly solid cancers such as breast cancer) [16]. All of these studies were based on existing medical or administrative records.

**People with genetic syndromes.** Fewer physical conditions were compared in the DS population than in the whole ID population. People with DS showed a higher risk of congenital malformations, hidradenitis suppurativa, and certain cancers. The well-established link between DS and congenital malformations was confirmed, especially for congenital heart defects (unspecified) (PR = 47, 95% CI = 44, 49; PR = 108, P$<10^{-6}$) [43, 78]. For specific heart defects, the risk ratios were very high, for example: 1009 (P$<10^{-6}$) for atrioventricular septal defect in a US infant population [78]; 850 (95% CI = 171, 1008) in a Norwegian population adjusted for maternal age and birth year [43]; and 510 (95% CI = 126.7, 999) in a South Korean population adjusted for age and sex [59]. There was also a significant increased risk of other malformations, such as gastrointestinal defects and congenital cataracts [78].

Regarding non-congenital disorders, one study reported a 5-times higher likelihood of hidradenitis suppurativa (OR = 5.24; 95% CI = 4.62, 5.94) [71]. All studies reported an elevated risk of leukemia in people with DS (SIR range of statistically significant estimates = 25.18 to 36). The highest risk was observed for acute myeloid leukemia (SIR range = 60 to 141) [42]. The high prevalence of leukemia in DS, especially in children, may account for the increased risk of cancer overall in children and adolescents in one study (SIR = 4.67; 95% CI = 1.9, 9.6) [116]. Otherwise, a non-significant decrease in overall cancer risk was reported in another study for people of all ages [117]. Lower risk of solid tumours has been observed in people with DS. One study reported a decreased risk of all solid cancers (SIR = 0.45; 95% CI = 0.34, 0.59) [117]. However, the evidence for individual solid cancers is inconclusive due to small case numbers [117].

For other genetic syndromes, evidence suggests people with Bardet-Biedl syndrome [120] or FXS [75] have a similar cancer risk to the general population.

## Discussion

This is the first systematic review to present a comprehensive summary of physical morbidity in people with ID from moderate to high quality population-based studies. We identified hitherto unknown health needs for this population, such as diabetes and asthma, and identified potential under-diagnosis of multiple disorders. Our findings strongly emphasize the importance of preventative health care, early detection, and close management of complex health care needs in people with ID. The substantial knowledge gap for many conditions and countries and the methodological limitations identified in the review offer guidance for future research directions.

### Main findings and comparison with other studies

This review identified high morbidity in people with ID. Epilepsy, cerebral palsy, sensory disorders, and metabolic and nutritional disorders were the most common conditions. Constipation, previously shown to be a leading health problem in this population [121], had only a moderately high estimated prevalence. The highest prevalence estimates in people with genetic syndromes

were reported for congenital heart defects and otitis media in children with DS and endocrine abnormalities in PWS. Unsurprisingly, most of the highly prevalent health conditions are associated with the disability or genetic syndrome [122]. Whilst our findings are primarily in keeping with the existing knowledge from previous reviews and clinical experience [11, 23, 33, 123–128], this review indicates their relevance in representative populations with ID.

Regarding comparisons to the general population, people with ID have increased risk of health conditions across 12 different ICD-10 disease chapters. While elevated risk has been previously documented for some conditions, such as epilepsy and sensory impairments [34, 23], our findings of associations between ID and diabetes, asthma, migraine, and hidradenitis suppurativa, are less well known [19, 20, 129, 130]. For people with DS, genetic susceptibility, along with associated obesity and abnormal immune system function, may predispose them to hidradenitis suppurativa [131, 132]. The other conditions are likely to occur more commonly among people with ID due to overrepresented risk factors [20, 133, 134].

The review also observed lower risks of some conditions in people with ID compared to the general population, including diverticular disorders, some circulatory diseases, and certain solid cancers. A lower risk of cancer and non-congenital circulatory diseases in this population is in line with mortality data [2, 3], possibly partly explained by low smoking rates [3], however, an alternative explanation is that many such diseases require proactive help-seeking behaviour and/or specific investigations. Thus, timely detection may be less likely in a person with ID [66, 135], for whom there are known barriers to seeking healthcare, especially to accessing preventative healthcare crucial to early disease detection [136]. All studies reporting lower disease risk relied on existing health records and thus may reflect an underestimation of morbidity, rather than a real reduction in risk.

Overall, the results of the current review need to be interpreted with caution for several reasons in addition to potential under-detection. First is the selection of the study population. Most samples were from a combination of disability and health services or primary health care registries in the UK, which may underrepresent people with mild ID, and may not be generalisable to other settings or populations. Relatedly, morbidity estimates of conditions linked to ID severity, such as epilepsy, may be impacted. Furthermore, few studies reported ID severity, limiting the clinical translation. Despite being robust, the estimated increased risk of several conditions was limited to adults with ID of any aetiology or DS only (for hidradenitis suppurativa), thus the generalisability to other age groups or other genetic syndromes is unknown.

## Implications for clinical and community practice

Our findings highlight the health needs specific to people with ID that require recognition and response. First is the high prevalence of several chronic health conditions, some of which may not be preventable but contribute to overall health burden and may compound management or outcomes associated with other diseases if left poorly managed. For example, epilepsy in people with ID is associated with more acute hospital visits, more comorbidities, and premature death [53, 137, 138]. Despite the importance of improving management for such conditions, current practice is generally suboptimal [139]. Second, this population appears to be more vulnerable to physical conditions which present risk for subsequent potentially avoidable deaths. Examples include diabetes, asthma, and injuries. This finding further emphasises the need for better prevention and tailored management of health conditions to reduce the risk. Last, optimal disease management cannot be achieved without early detection. The under-detection implied by this review suggests inadequate inclusion of people with ID in screening and preventative health programs and highlights the urgent need for this to be remedied. For example, evidence-based tools such

as the Comprehensive Health Assessment Program can be introduced to help clinical decision making when attending to this group [140, 141]. Additionally, adjustments to services are needed to cater for needs related to disability features, such as immobility, and those related to other common concomitant disabilities.

## Implications for future research

We observed two major knowledge gaps. The first is the lack of research on some age- and lifestyle-related health conditions in people with ID. This is concerning as the morbidity rates and related burdens are likely to increase as a result of increasing longevity and generally unhealthier lifestyles associated with transitions to living in the community [142–145]. Most evidence for cardiovascular disorders and risk factors stems from the UK [10, 68, 81, 97, 112], thus the generalisability to other countries is unclear. Future research should aim to provide representative data from different countries for health conditions related to aging and lifestyle factors, especially from low- and middle-income settings, where people with ID may be less likely to have their health needs identified and met [146, 147].

As discussed previously, despite being relatively robust, the current evidence may be subject to selection bias. It is important to maximise the representativeness of future study populations. Linkage of multiple administrative and clinical datasets, including multiple levels of healthcare and disability services, may be one solution [148], as illustrated in recent high-quality studies [40, 118]. Data linkage studies should also capture disability severity to enhance the clinical utility of their results. One intrinsic limitation of this approach is that detection bias cannot be avoided without the inclusion of health screening programs data.

Finally, most studies reported only prevalence estimates of health problems in people with ID. More longitudinal studies are needed to estimate incidence rates to identify causal links more robustly between ID and health conditions.

## Strengths and limitations of the current review

A major strength of this review is that we extracted individual health conditions rather than broad categories (e.g., cardiovascular disorders). Given that disorders belonging to one broad disease group may have different morbidity patterns in people with ID (e.g., heart failure and hypertension), information based on broad disease groups can be misleading. Another strength is our approach to include population-based cohorts, so samples identified solely from healthcare services, except the UK GP sample or birth cohorts, were not included. Although we risked excluding high-quality clinical studies, we avoided including potentially biased samples which presented due to ill-health. Finally, we implemented a rigorous quality appraisal and excluded low-quality studies from the evidence synthesis.

There are some limitations of our review. First, excluding studies with voluntary and hospital-based samples reduced the representation of people with rare genetic syndromes. Second, we could not consider the diagnostic classification systems in the evidence synthesis, as this information was unavailable in many studies. Third, the identification of risk estimates of physical health conditions may be incomplete, as risk was not our primary outcome and was not included in the search strategy. Finally, although the adapted NOS we used for quality assessment has not been validated, our adaptations only optimised the wording and categories to incorporate ID and match our study design. The NOS has been found to have poor agreement between reviewers, and its scores depended on the extent to which the research methods were reported [149, 150]. However, any disagreements between reviewers were resolved through discussion.

## Conclusions

The wide range of physical health conditions associated with ID indicates the complexity of health needs of this disadvantaged group. A comprehensive picture of morbidities can assist people with disability and their health professionals, disability professionals, and families to pursue timely diagnosis and better disease management. It will also assist services and governments to develop responsive health policy and health promotion programs. Age- and lifestyle-related physical conditions are worthy of more attention in future research. Globally, increased awareness and targeted preventative health care initiatives may improve the health outcomes for this vulnerable population.

## Supporting information

**S1 File. Adapted Newcastle-Ottawa Scale (NOS).**
(DOCX)

**S2 File. Characteristics of included studies.**
(DOCX)

**S3 File. Study quality assessment scores.**
(DOCX)

**S4 File. Results tables, including prevalence, incidence, and risk estimates.**
(DOCX)

**S5 File. Search terms and steps.**
(DOCX)

**S1 Checklist.**
(DOC)

## Acknowledgments

We thank Dr. Rachael Cvejic (Department of Developmental Disability Neuropsychiatry (3DN, UNSW) for her input in designing how the diseases should be grouped and mapped to the ICD-10 disease chapters.

## Author Contributions

**Conceptualization:** Peiwen Liao, Claire Vajdic, Julian Trollor, Simone Reppermund.

**Formal analysis:** Peiwen Liao.

**Funding acquisition:** Claire Vajdic, Julian Trollor, Simone Reppermund.

**Investigation:** Peiwen Liao, Simone Reppermund.

**Methodology:** Peiwen Liao, Claire Vajdic, Julian Trollor, Simone Reppermund.

**Supervision:** Claire Vajdic, Julian Trollor, Simone Reppermund.

**Writing – original draft:** Peiwen Liao.

**Writing – review & editing:** Claire Vajdic, Julian Trollor, Simone Reppermund.

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
