## [Decision Letter · Decision Letter 0]

5 May 2021

PONE-D-21-05937

Prevalence and incidence of physical health conditions in people with intellectual disability – a systematic review

PLOS ONE

Dear Dr. Liao,

Thank you for submitting your manuscript to PLOS ONE. After careful consideration, we feel that it has merit but does not fully meet PLOS ONE’s publication criteria as it currently stands. Therefore, we invite you to submit a revised version of the manuscript that addresses the points raised during the review process.

Please address the final reviewer suggestions.

We look forward to receiving your revised manuscript.

Kind regards,

Shahrad Taheri

Academic Editor

PLOS ONE

Journal Requirements:

3. We note that your literature search was performed on February 2020; to allow an up-to-date view of the topic, we would request that the search is updated.

Moreover, please include in the main text a Table showing in more detail the results of the quality assessment (reporting how each included study scored on every item of the scale); and please discuss whether the modified NOS scale was previously validated.

Reviewers' comments:

Reviewer's Responses to Questions

**Comments to the Author**

1. Is the manuscript technically sound, and do the data support the conclusions?

Reviewer #1: Yes

Reviewer #2: Yes

2. Has the statistical analysis been performed appropriately and rigorously? 

Reviewer #1: Yes

Reviewer #2: Yes

3. Have the authors made all data underlying the findings in their manuscript fully available?

Reviewer #1: Yes

Reviewer #2: Yes

4. Is the manuscript presented in an intelligible fashion and written in standard English?

Reviewer #1: Yes

Reviewer #2: Yes

5. Review Comments to the Author

Reviewer #1: First, I want to thank the authors for writing such a wonderful and interesting paper which examined the prevalence of physical health conditions in individuals with intellectual disabilities. This is an incredibly important topic and I feel this paper will greatly add to the body of work in this area. That being said, I had a few small comments with regards to the methodology, specifically around your search terms and your inclusion dates.

Overall, I enjoyed reading this article and as you can see by the few comments I have, was engrossed in your research and found few flaws. Please see my more specific comments below. I hope that you find them helpful, as I believe they will improve the quality of what is already a high quality piece of work.

INTRODUCTION

I would suggest the author starts by providing an introduction to what an intellectual disability is. Not all readers of PLOS are familiar with IDs and an introduction and overview of them may be helpful.

Line 47 - Curious why the authors did not mention the prevalence of cardiovascular conditions in this population. There is extensive literature in this area.

Line 51 – I would provide an example of what you mean by single health conditions

Line 54 – The authors should explain what they mean by low quality studies as not all readers would be familiar

METHODS

I have some concern over two pieces of the methodology.

First, the search terms could be broadened to ensure all relevant articles were reviewed. First, you used the term intellectual disability or ID – in the UK an intellectual disability is referred to as a learning disability, how did your search terms account for this. Also does physical disorder capture different, diseases and conditions?

Second – your search is over a year old. This should be updated.

Line 74 – Provide an example of genetic syndromes

Line 83 – Why were studies with voluntary participants excluded?

RESULTS

Line 144 – predominantly mild to profound. Isn’t that the entire breadth of what we consider to be ID’s? Was it more mild diagnoses? Please clarify.

DISCUSSION

I have no comments for the authors as this section was very well written and a joy to read.

Reviewer #2: I think is a very thorough and professional piece of work. All the components of a systematic review are there. The authors seem experience in systematic reviews.

It a very good piece of work and needed to fill gaps in knowledge. Would be a valuable contribution.

Just one note: I am not sure if the studies that had subjects with CP used a classification system for ID? Just making sure the subjects all had ID on top of CP. Although you did address this in your limitations.

6. PLOS authors have the option to publish the peer review history of their article (what does this mean?). If published, this will include your full peer review and any attached files.

Reviewer #1: No

Reviewer #2: **Yes: **Portia Ho

---

## [Author Response · Author response to Decision Letter 0]

29 Jun 2021

Response to reviewer comments

We would like to thank the reviewers for their helpful and constructive comments. Responses to each reviewer’s comments are listed below. 

Reviewer 1:

Comment: I would suggest the author starts by providing an introduction to what an intellectual disability is. Not all readers of PLOS are familiar with IDs and an introduction and overview of them may be helpful.

Response: Thank you for raising this issue. We agree and have added a brief introduction to the introduction:

Page 3, line 41-42: “People with intellectual disability (ID), defined by cognitive and adaptive-functioning impairments with onset during childhood, constitute about 1% of the global population”

Comment: Line 47 - Curious why the authors did not mention the prevalence of cardiovascular conditions in this population. There is extensive literature in this area.

Response: We agree that there are many publications about cardiovascular conditions in people with ID. However, we did not include them in the introduction because the current evidence is inconclusive. Although some studies have suggested that people with ID may have increased risk of cardiovascular disorders [1], others observed a lower risk of certain cardiovascular disorders (as shown in our review). The variation is highly likely due to differences in study populations, study design, and how disease was ascertained and classified. Furthermore, we were unable to find a good synthesis of the evidence for the whole population with ID, making this issue one of the knowledge gaps that needed to be addressed in this review. In light of the reasons above, we did not choose to present information about cardiovascular conditions.

Comment: Line 51 – I would provide an example of what you mean by single health conditions

Response: Thank you for the suggestion, we have added an example on Page 4, line 57: “Previous reviews mainly examined single health conditions [11, 18-32], such as epilepsy, or a specific sub-population (e.g., people with Down Syndrome (DS)) [15, 33].”.

Comment: Line 54 – The authors should explain what they mean by low quality studies as not all readers would be familiar

Response: Thank you for raising this question. In this case, we are referring to studies likely subject to bias, including selection bias, misinformation bias, and so on. To clarify this, we explain what low quality means in the following paragraph. 

Page 4, line 59-61: “Importantly, few reviews required representative populations [35], and none, to our knowledge, excluded studies potentially subject to bias.”

Comment: I have some concern over two pieces of the methodology.

First, the search terms could be broadened to ensure all relevant articles were reviewed. First, you used the term intellectual disability or ID – in the UK an intellectual disability is referred to as a learning disability, how did your search terms account for this. Also does physical disorder capture different, diseases and conditions?

Response: We confirm that we did include learning disability as a separate search term, as described in the supporting information S13. We also confirm that physical disorders capture different diseases and conditions according to the search strategy. We used Medical Subject Headings (MeSH) in the search, and the MeSH term for physical health conditions includes different diseases. Further details are listed in the supplementary file S13. 

Comment: Second – your search is over a year old. This should be updated.

Response: We agree and have updated the search. We identified another 13 eligible studies, 5 of which were of moderate to high quality and included in the final outcome analyses. The characteristics of the 13 studies were added to Tables S4-7, and the outcomes from the 5 studies were added to Tables S9-12. The related information (e.g., the number of studies) in the main manuscript were updated, and we also updated Figures 1, 2 and 3 accordingly. The main changes in the manuscript are listed below:

• New estimate for incidence of celiac disease in people with DS (Reference No. 61: Ostermaier et al. (2020)): 

Page 12, line 243 – 246: “Two studies reported the incidence of other conditions. A British study examined injury incidence and found that on average 20 per 100 adults with ID experienced accidental injuries within one year [44]. Another study reported an incidence of 32.5 per 10 000 person-years for celiac disease in people with DS [61].”

• New estimate for relative risk of congenital heart defects in people with DS (Reference No.59: Cho et al. (2020)): 

Page 14, line 295 – 296: “…and 510 (95% CI=126.7, 999) in a South Korean population adjusted for age and sex [59].”

Comment: Line 74 – Provide an example of genetic syndromes

Response: Thank you for this suggestion, we have added an example. 

Page 5, line 81: “… in people with ID or genetic syndromes invariably related to ID, such as DS.”

Comment: Line 83 – Why were studies with voluntary participants excluded?

Response: In this review, we were aiming to synthesise the evidence based on population-based samples of people with ID, to present results that are as generalisable as possible. Voluntary or convenience sampling does not fulfill this requirement. The participants sampled by these methods are usually self-selected, and the data obtained may not represent the target population [2-4]. 

To clarify this, we have made some edits in the text: Page 5, line 91-93: “Studies with voluntary/convenience participants or people with additional specific health conditions were excluded, as they were not likely to represent the whole ID population and were at risk of selection bias.”

Comment: Line 144 – predominantly mild to profound. Isn’t that the entire breadth of what we consider to be ID’s? Was it more mild diagnoses? Please clarify.

Response: We agree and have reworded the sentence to avoid confusion. 

Page 8, line 158-159: “Twenty-two studies reported ID severity, and most covered all severities, from mild to profound”. 

Comment: I have no comments for the authors as this section was very well written and a joy to read.

Response: We are grateful to the reviewer for this feedback.

Reviewer 2:

Comment: I think is a very thorough and professional piece of work. All the components of a systematic review are there. The authors seem experience in systematic reviews. It a very good piece of work and needed to fill gaps in knowledge. Would be a valuable contribution. I am not sure if the studies that had subjects with CP used a classification system for ID? Just making sure the subjects all had ID on top of CP. Although you did address this in your limitations.

Response: We would like to thank Reviewer 2 for the positive comments. To answer the question: we did not include subjects with CP if they did not have a diagnosis of ID.

Response to the journal requirements:

Response: Thank you. We confirm that the manuscript meets PLOS ONE’s style requirement. 

Response: We confirm that no studies that are cited in the review have been retracted. After the updated search, we included another 13 studies in the review, the full list of the citations of these studies are in the supplementary file S4 and S5 and S6 and S7_Table. We also included 2 references about NOS (Page 20, line 425), and added one reference which was missing in the last version (Page 10, line 208). Citations added to the main manuscript are summarised below:

• No. 58 Bishop L et al., 2020

• No. 59: Cho WK et al., 2020

• No. 60: Kristianslund O et al., 2021

• No. 61: Ostermaier KK et al., 2020

• No. 103: Tedeschi AS et al., 2015

• No. 119: Cuypers M et al., 2021

• No. 151: Luchini C et al., 2017

• No. 152: Lo CK et al., 2014

Due to changes in the quality assessment scores, one citation (Baccichetti et al., 1990) has been removed from the main manuscript. Please see the justification for this below in the section “Other changes made by the authors”. 

3. We note that your literature search was performed on February 2020; to allow an up-to-date view of the topic, we would request that the search is updated.

Response: The search was updated on 11 May 2021. The details are outlined in our response to Reviewer 1. 

4. Moreover, please include in the main text a Table showing in more detail the results of the quality assessment (reporting how each included study scored on every item of the scale); and please discuss whether the modified NOS scale was previously validated.

Response: The new table is attached (S8_Table), and the relevant discussion has been added to the manuscript:

Page 20, line 421-426: “Finally, although the adapted NOS we used for quality assessment has not been validated, our adaptations only optimised the wording and categories to incorporate ID and match our study design. The NOS has been found to have poor agreement between reviewers, and its scores depended on the extent to which the research methods were reported [151, 152]. However, any disagreements between reviewers were resolved through discussion.” 

On account of the size of this table, we would strongly prefer to add it to the supporting information as a supplementary file and have included it as such pending your further review. 

We also updated the supplementary table numbers since this table became Table S8. The previous Table S8 became S9, S9 became S10, S10 became S11, S11 became S12, and the file S12 became S13. The updates were made in the relevant supplementary files and file names, and in the section of “Supporting information” at the end of the main manuscript (Page 30-31).

Others changes made by the authors:

During the revision process, we identified a few minor mistakes in our quality assessment scores, and we also revised a score for one study according to the information ascertained during the updated search (Burke et al. (2019)). This resulted in some estimates of morbidity rates (from 3 studies) and of relative risk (from 1 study) falling into the category of low quality, and therefore being excluded from the final evidence synthesis. These changes, along with the newly added studies from the updated search, do not change our results or conclusion. Please find the details below:

In addition to Burke et al. (2019), four other studies were involved: Kapell et al. (1998), Baccichetti et al. (1990), Prasher et al. (1995), and Warburg (2001). The change in the quality score can be tracked in the Supplementary file S4 and S5 and S6 and S7_Table (Table S4, S5, and S7). 

The removal of these estimates resulted in the following edits in the manuscript:

• The removal of relative risk estimates of one study (Kapell et al. (1998), reference No. 95) resulted in the following edits:

Page 14, line 289-290: “..thyroid disorders…” and line 300-302: “…and another reported a 12.5 times higher risk for thyroid disorders among people with DS (age-standardised morbidity ratio=12.5; 95%CI=9.1-16.8)” were deleted, as this study provided the only estimate of relative risk of thyroid disorders in people with DS in the final synthesis stage. 

Page 15, line 311-314: these two sentences were deleted due to the same reason that this study provided the only estimates for the outcome analysis. 

• The removal of prevalence estimates of one study (Baccichetti et al. (1990)) resulted in the following edits:

Page 10, line 209: “…congenital heart defects (unspecified) (14.1%-79.2%)”.

Page 10, line 211: “…common skin diseases (13.0%-23.4%)”. 

Page 10, line 212-213: “…lower respiratory tract infection (11.4%-27.0%) [97, 99]”

This is because the removal of these estimates affected the range of the prevalence estimates. 

The above, together with the removal of the estimates from the other two studies (Prasher et al. (1995), and Warburg (2001)) caused changes to the Supplementary file S9 and S10 and S11 and S12_Table and figures 1, 2, and 3. 

Other minor edits:

Page 3, line 50-52: “Compared to the general population, people with ID have a higher prevalence of physical conditions [10], especially neurological disorders, sensory impairments, obesity, and constipation, and congenital malformation”. We added “and congenital malformation”, an oversight from our original draft. 

Page 6, line 104-107: “We adapted the Newcastle-Ottawa Scale (NOS), a validated and widely used tool, to assess the quality of included studies (table S1 and S2) [37]. The NOS framework was unchanged, our adaptations ensured the NOS was fit-for-purpose for our study design.” We added a more detailed description of the NOS tool and explained how we adapted the tool to fit our study design. 

Page 10, line 207: “…visual impairment including blindness (0.8%-34.9%) [94-96]…”. And Page 10-11, line 213-218: “For hearing and visual impairments, the lower prevalence estimates were found for specific hearing [98, 100, 102, 104] or visual impairments [94, 96], including deafness (0.9%) [101] and blindness (0.8%) [94], and higher prevalence estimates for unspecified impairments. The lowest prevalence of unspecified hearing loss in children with DS was based on parent-reported data.” The reason for these two edits is that we did not include blindness in the main body of the manuscript due to its low prevalence, but we have now added it because it belongs to visual impairments. This change is also reflected in the figure 3. 

References:

1. Draheim CC. Cardiovascular disease prevalence and risk factors of persons with mental retardation. Ment Retard Dev Disabil Res Rev. 2006;12(1):3-12. doi: 10.1002/mrdd.20095. 

2. Australian Bureau of Statistics. Sample Design. Access date: 15 June 2021. URL: https://www.abs.gov.au/websitedbs/d3310114.nsf/home/Basic+Survey+Design+-+Sample+Design

3. Bornstein MH, Jager J, Putnick DL. Sampling in Developmental Science: Situations, Shortcomings, Solutions, and Standards. Dev Rev. 2013;33(4):357-370. doi:10.1016/j.dr.2013.08.003

4. Elfil M, Negida A. Sampling methods in Clinical Research; an Educational Review. Emerg (Tehran). 2017;5(1):e52.

---

## [Decision Letter · Decision Letter 1]

4 Aug 2021

Prevalence and incidence of physical health conditions in people with intellectual disability – a systematic review

PONE-D-21-05937R1

Dear Dr. Liao,

We’re pleased to inform you that your manuscript has been judged scientifically suitable for publication and will be formally accepted for publication once it meets all outstanding technical requirements.

Kind regards,

Shahrad Taheri

Academic Editor

PLOS ONE

Additional Editor Comments (optional):

Reviewers' comments:

Reviewer's Responses to Questions

**Comments to the Author**

1. If the authors have adequately addressed your comments raised in a previous round of review and you feel that this manuscript is now acceptable for publication, you may indicate that here to bypass the “Comments to the Author” section, enter your conflict of interest statement in the “Confidential to Editor” section, and submit your "Accept" recommendation.

Reviewer #1: All comments have been addressed

Reviewer #2: All comments have been addressed

2. Is the manuscript technically sound, and do the data support the conclusions?

Reviewer #1: Yes

Reviewer #2: Yes

3. Has the statistical analysis been performed appropriately and rigorously? 

Reviewer #1: Yes

Reviewer #2: I Don't Know

4. Have the authors made all data underlying the findings in their manuscript fully available?

Reviewer #1: Yes

Reviewer #2: Yes

5. Is the manuscript presented in an intelligible fashion and written in standard English?

Reviewer #1: Yes

Reviewer #2: Yes

6. Review Comments to the Author

Reviewer #1: Thank you for addressing my comments and concerns. I want to thank you again for giving me the opportunity to review such an excellent paper. Well done.

Reviewer #2: I am satisfied with the responses. The changes are good and very thorough. From my limited experience, the data analysis read logically but I am not an expert to comment on that area.

7. PLOS authors have the option to publish the peer review history of their article (what does this mean?). If published, this will include your full peer review and any attached files.

Reviewer #1: No

Reviewer #2: **Yes: **Portia Ho

---

## [Editor Report · Acceptance letter]

9 Aug 2021

PONE-D-21-05937R1 

Prevalence and incidence of physical health conditions in people with intellectual disability – a systematic review 

Dear Dr. Liao:

I'm pleased to inform you that your manuscript has been deemed suitable for publication in PLOS ONE. Congratulations! Your manuscript is now with our production department. 

Kind regards, 

on behalf of

Dr. Shahrad Taheri 

Academic Editor

PLOS ONE